# Optimization of high-channel count, switch matrices for multinuclear, high-field MRI

**Jörg Felder**[1]*, **Chang-Hoon Choi**[1], **Yunkyoung Ko**[1], **N. Jon Shah**[1,2,3,4]

**1** Institute of Neuroscience and Medicine -4, Forschungszentrum Jülich, Jülich, Germany, **2** Institute of Neuroscience and Medicine -11, Forschungszentrum Jülich, Jülich, Germany, **3** JARA—BRAIN—Translational Medicine, Aachen, Germany, **4** Department of Neurology, RWTH Aachen University, Aachen, Germany

* j.felder@fz-juelich.de

## Abstract

Modern magnetic resonance imaging systems are equipped with a large number of receive connectors in order to optimally support a large field-of-view and/or high acceleration in parallel imaging using high-channel count, phased array coils. Given that the MR system is equipped with a limited number of digitizing receivers and in order to support operation of multinuclear coil arrays, these connectors need to be flexibly routed to the receiver outside the RF shielded examination room. However, for a number of practical, economic and safety reasons, it is better to only route a subset of the connectors. This is usually accomplished with the use of switch matrices. These exist in a variety of topologies and differ in routing flexibility and technological implementation. A highly flexible implementation is a crossbar topology that allows to any one input to be routed to any one output and can use single PIN diodes as active elements. However, in this configuration, long open-ended transmission lines can potentially remain connected to the signal path leading to high transmission losses. Thus, especially for high-field systems compensation mechanisms are required to remove the effects of open-ended transmission line stubs. The selection of a limited number of lumped element reactance values to compensate for the for the effect of transmission line stubs in large-scale switch matrices capable of supporting multi-nuclear operation is non-trivial and is a combinatorial problem of high order. Here, we demonstrate the use of meta-heuristic approaches to optimize the circuit design of these matrices that additionally carry out the optimization of distances between the parallel transmission lines. For a matrix with 128 inputs and 64 outputs a realization is proposed that displays a worst-case insertion loss of 3.8 dB.

## Introduction

Modern magnetic resonance imaging (MRI) systems rely heavily on parallel imaging techniques using multi-channel, phased array coils. Currently, they are routinely employed to improve image quality, to reduce the scan time or to obtain more information (e.g. in multi-nuclear or multi-parametric measurements) within a given time frame. Moreover, reducing

**Data Availability Statement:** All relevant data are within the manuscript and its Supporting Information files. We have also uploaded all source code of the project on the public Gitlab repository of the Research Center Jülich where it is freely

available (https://gitlab.fz-juelich.de/j.felder/multinuclear-switch-matrix).

**Funding:** The author(s) received no specific funding for this work.

**Competing interests:** NJS and JF are co-founders of Affinity Imaging GmbH a spin-off company that manufactures high field MRI coils for research purposes. This does not alter our adherence to PLOS ONE policies on sharing data and materials.

the scan time brings about further benefits in terms of increasing patient compliance (shorter scan times are more comfortable for the patient) and enables an increased workload of expensive MRI machinery (patient throughput can be increased). In fact, several applications would not be feasible at all without parallel imaging techniques. These include, but are not limited to, imaging the beating heart [1, 2], fast functional/diffusion imaging [3–7] and correcting unwanted artifacts, e.g. aliasing and ghosting [8, 9].

With the increase in the number of elements in phased array coils [10–12], more and more parallel receive channels are required per MRI system. Unfortunately, this has a significant negative impact in terms of machine cost due to the requirement for additional receiver units and a technologically complex patient table [13], on patient comfort and operator workflow arising from the bulky cable bundles on the local RF coils [13], and on the large amounts of data that require handling prior to final image reconstruction. Thus, the number of receive antennas (coils)–or at least the number of physical connection points in the patient table [14]– present in an MRI examination frequently exceeds the number of receivers available in the system. The problem is aggravated when large fields-of-view are to be covered and multiple coil arrays are used during a scan, e.g. in spine imaging and moving table acquisitions.

An early commercial solution to the problem was the incorporation of mode matrices in the receiver coil array [15]. These make use of the fact that the majority of the receive signal's power is contained in a small number of RF modes. The concept can be compared to the use of circular polarized receive coils which employ two quadrature coil elements and combines these into a single receive signal. Thus, high-channel-count coil systems can be connected to fewer receiver channels, while avoiding significant penalties in the receive signal-to-noise ratio (SNR). However, the mode matrix approach significantly reduces the available acceleration factor for a given number of receive elements as it actually reduces the physical number of receive elements [16]. This factor is particularly significant in high-field systems, which are increasing in availability, as they potentially allow higher speed-up factors [17]. Thus, most modern MRI scanners have a large number of physical coil connectors distributed on the patient table that need to be flexibility connected to the available receivers with the established routing depending on the coils selected for the desired imaging task e.g. when full body coverage is desired [14] or multiple different type of local coils are employed in a single imaging session [13].

Signal routing from the physical coil connectors commonly located on the patient table to the receivers located in the technical room outside the Faraday shield in clinical MRI systems is usually achieved by using a module termed either the "switch matrix" (a term originally used in telecommunications and computer communication networks (e.g. [18] or "matrix switch" (e.g. [19]. The switch matrix is commonly located close to the magnet to avoid long coaxial cable bundles but physically behind the initial low noise preamplifier in order to maintain the highest possible SNR. A large number of implementations are feasible and can generally be categorized as crossbar switches or multiplex matrix switches [20]. Both of these, as well as hybrid topologies, have been used in MRI applications [21, 22]. Crossbar switch matrix implementations can either be achieved with the use of integrated circuits configured as single-pole double-throw (SPDT) switches, e.g. based on GaAs, GaN [23], or MEMS [24] switches, or by employing simpler single-pole single-throw (SPST) configurations, e.g. by using PIN-diodes. While the former implementation seems to be technologically more robust, as it allows to disconnect the remaining transmission line length from the active signal path and therefore does not contain remaining physically connected open-ended transmission lines, it requires the receive signal to pass through a high number of active switches. This potentially leads to high signal attenuation. As an example, a well-designed GaAs IC with suitably high linear power handling capability for MRI receive signals easily displays an insertion loss of 0.35 dB (e.g.

SKY13351-378LF–Skyworks Solutions, Inc., USA). The receive signal has to cross this device (M-1) x N times in the worst case with M being the number of input lines and N the number of output lines.

Recently, PIN-diode controlled switch matrices have been shown to be feasible even in high-field MRI applications by either compensating open transmission line impedances with suitable lumped element terminations in one direction [25] or in both directions [26]. In addition, a method to reduce the PCB footprint for large size matrices using a combination of switch types has been presented [27].

To date, switch matrix design has mostly been concerned with routing the signals of the proton Larmor frequency as the number of available high receive channel arrays operating at non-proton frequencies at clinical field strength is limited to research coil implementations, e.g. [28, 29]. However, the routing of X-nuclei (non-proton, resonating at different frequencies) signals might still be required if the coil connectors on the patient table do not provide dedicated plugs for the X-nucleus' receive coil arrays.

A number of switch matrix implementations with variable degrees of flexibility in routing configuration have been described in the literature: sparse or cascaded matrix designs are described in [13, 30, 31] and a full switch matrix is described in [32]. While they are routinely employed in MRI scanners operating at clinical field strength of 1.5 T and 3 T, in some commercial 9.4 T and 7 T systems, the switch matrix was removed and the signal was routed directly without the possibility to switch it to different receiver units. However, with the increased availability of high-field scanners, the benefits of X-nuclei imaging is likely to attract increased interest [33, 34]. As multichannel, X-nucleus coil arrays also provide improved SNR compared to volume coils, as is also the case in proton imaging, there is compelling motivation to also use coil arrays in these SNR-starved applications. In fact, multichannel X-nucleus arrays have been under active development for a number of years, and one can compare exemplary implementations in [35, 36] or [37] for a review on the topic. In this context, designing a switch matrix suitable for operation with protons and the most common X-nuclei at high-field is desirable. It would, for example, enable the combined operation of a CP mode proton coil with a 32-channel sodium array in a system equipped with 32 receive channels or the connection of more advanced double-tuned arrays as described in [38]. Note, that switch matrix implementations operating at the proton frequency as well as at one or more X-nucleus frequency has been previously described in [39].

In this manuscript we present a novel design methodology for a low-loss, high channel-count switch matrix based on SPST switches employing PIN diodes [26] capable of operating at, for example, 400 MHz (proton - $^{1}$H), 376 MHz (Fluorine - $^{19}$F), 162 MHz (Phosphorus - $^{31}$P), 106 MHz (Sodium - $^{23}$Na) and 54 MHz (Oxygen - $^{17}$O), these being the resonance frequencies of the said nuclei on a 9.4 T MR system. The method is based on multi-parameter optimization strategies and uses analytical transmission line formulas to derive optimum transmission line spacing and compensation elements on both horizontal and vertical switch matrix transmission lines.

## Methods

Fig 1 shows the topology of a switch matrix employing a single PIN-diode at each junction. Note that single PIN-diode switches can make a connection between an input and an output, but cannot break it–they are acting in a SPST manner. Therefore, the transmission line stubs remain electrically connected. Thus, attenuation of the signal occurs, with the level of attenuation depending on the impedance transformation of the open transmission line to the location where the PIN-diode connects upper and lower transmission lines. The remaining vertical and

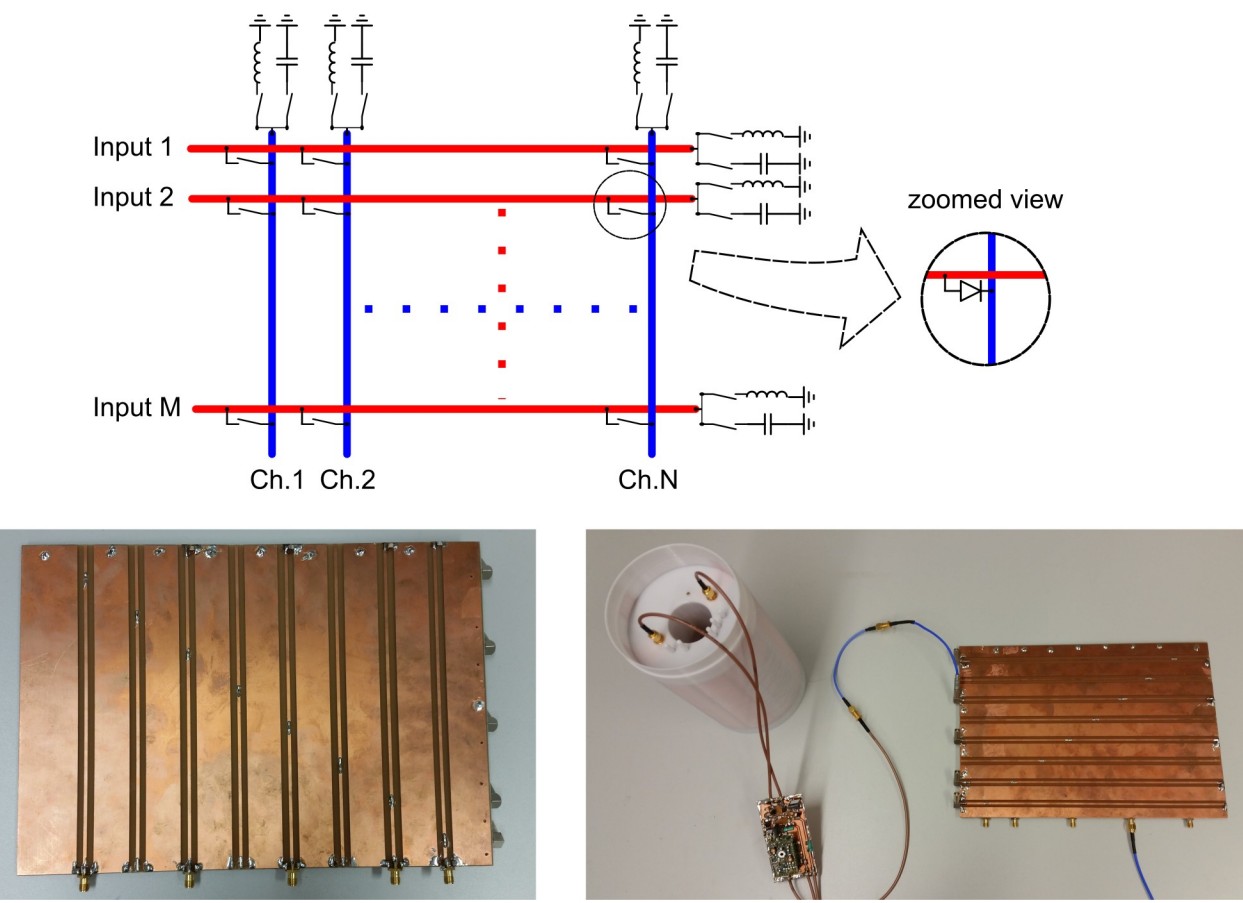

**Fig 1. Topology of a crossbar switch matrix (top) and prototype board (bottom).** The top image shows an M-input x N-output freely configurable crossbar switch matrix using SPST switches. Selectable compensation elements, e.g. one capacitor and one inductor are located at the end of each transmission line stub. The SPST switches can be realized as a single PIN diode. The bottom images show the prototype PCB as well as the prototype connected to a birdcage coil including preamplifier and transmit/receive switch.

horizontal stub lines will contribute to the signal attenuation across the switch matrix as described in [26].

In order to demonstrate the implementation challenges, the attenuation arising from the unterminated transmission lines for a low channel-count (5 inputs x 5 outputs) switch matrix has been measured on a prototype board [26] for the connections shown in color in Fig 2 for a number of nuclei frequently used in high field MRI. Even for the small-sized matrix, the task of determining suitable compensation elements is tedious and evaluation of optimal compensation element values quickly becomes infeasible. Thus, for the sake of simplicity, Fig 2 reports results obtained for a single compensation element of 91 pF. In this prototype "switching" from unterminated to terminated stub lines was accomplished by soldering the termination impedance on the PCB. Only a limited number of switch matrix configurations with different stub line lengths were evaluated experimentally in order to demonstrate the negative influence of unterminated transmission lines on the IL per se. In a product implementation connection of the designated termination resistance will be implemented via SPST PIN diode switches.

The worst-case IL in the small laboratory sample matrix evaluated in Fig 2 is 24 dB in the uncompensated case, while it is only 1.94 dB for a connection from input #1 to output channel #5 on the proton frequency when compensation elements are employed. To investigate the influence of IL on SNR and in order to establish the requirements for the circuit, optimization experiments

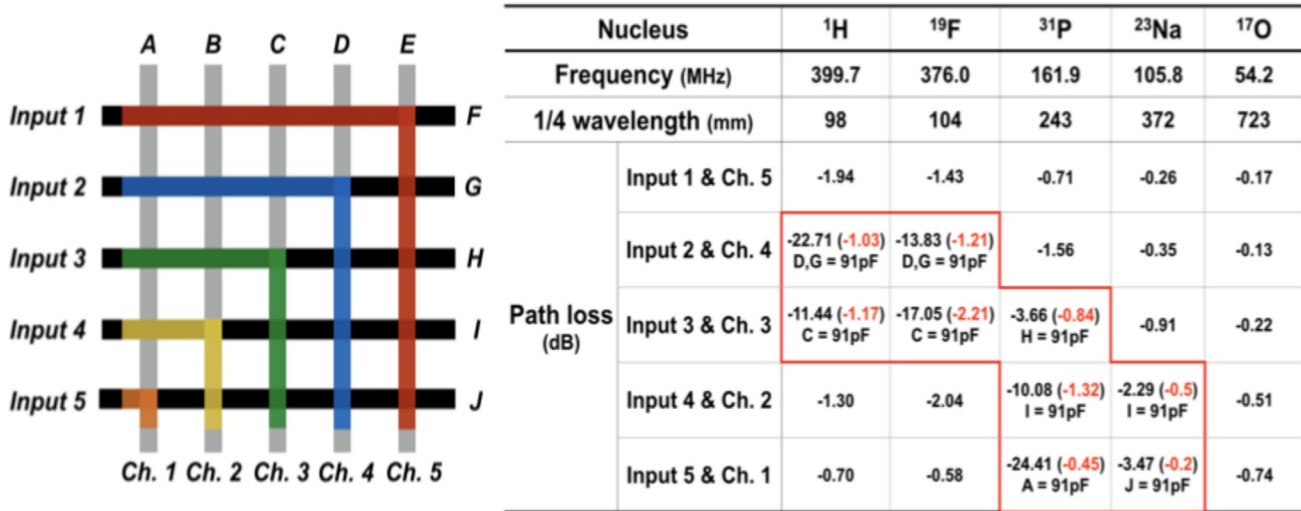

| Nucleus | | ¹H | ¹⁹F | ³¹P | ²³Na | ¹⁷O |
|---|---|---|---|---|---|---|
| Frequency (MHz) | | 399.7 | 376.0 | 161.9 | 105.8 | 54.2 |
| 1/4 wavelength (mm) | | 98 | 104 | 243 | 372 | 723 |
| Path loss (dB) | Input 1 & Ch. 5 | -1.94 | -1.43 | -0.71 | -0.26 | -0.17 |
| | Input 2 & Ch. 4 | -22.71 (-1.03) D,G = 91pF | -13.83 (-1.21) D,G = 91pF | -1.56 | -0.35 | -0.13 |
| | Input 3 & Ch. 3 | -11.44 (-1.17) C = 91pF | -17.05 (-2.21) C = 91pF | -3.66 (-0.84) H = 91pF | -0.91 | -0.22 |
| | Input 4 & Ch. 2 | -1.30 | -2.04 | -10.08 (-1.32) I = 91pF | -2.29 (-0.5) I = 91pF | -0.51 |
| | Input 5 & Ch. 1 | -0.70 | -0.58 | -24.41 (-0.45) A = 91pF | -3.47 (-0.2) J = 91pF | -0.74 |

**Fig 2. Signal routes (colored) evaluated at different frequencies (left) and insertion loss measured on the bench for the different nuclei considered here.** If the insertion loss was above 2 dB, a single value compensation (91 pF) was attached at one or both transmission lines and the insertion loss re-measured for all possible combination of compensation elements. All values are in dB; all input lines are designated as "Input" and all output lines as "Channel" with consecutive numbering. Transmission line stubs are labeled with alphabetical capital letters A to J. For the sake of clarity, only a limited number of matrix configurations have been evaluated experimentally, e.g. the switch was configured with fixed connections from input #1 to output #5, input #2 to output #4 and so forth to demonstrate the effect of transmission line stubs of different length for different resonant frequencies. In the table on the right, the value for the insertion loss shown in black indicates values measured for the unterminated case (that is, without the 91 pF attached to the transmission line stub). Red values are the ones representing IL for the terminated case (where the 91 pF are connected to the transmission line stubs). For the terminated case, the location of the termination capacitance is given underneath the measured IL values.

were carried out on a 9.4 T animal scanner [40] by inserting a variable attenuator at the location of the proposed switch matrix. Single slice gradient echo (TE = 10 ms, TR = 100 ms, FA = 15) images were acquired using a linear-polarized birdcage resonator and a 63 ml sample (per 1000 g distilled water 5 g NaCl and 1.25 g $NiSo_4$ x 6 $H_2O$) to measure SNR as a function of attenuation as described in [41], Method 2. The results of the measurement are given in Fig 3.

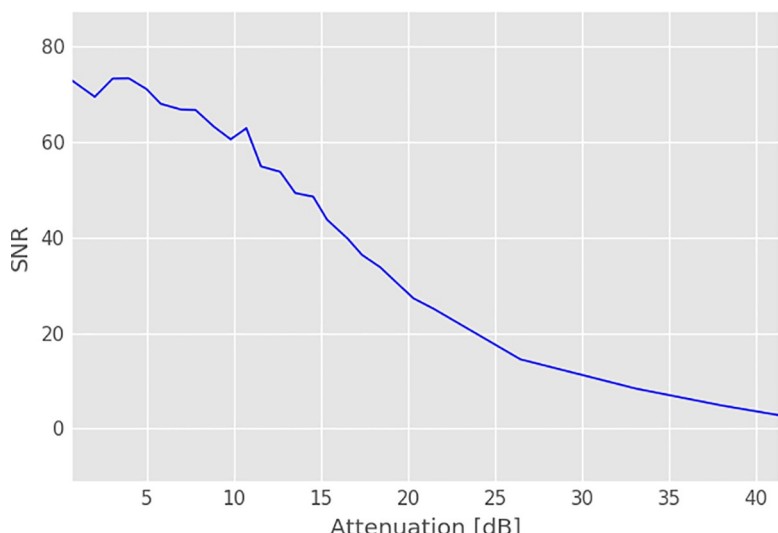

**Fig 3. Measured SNR of a GRE image versus signal attenuation at the position of the proposed switch matrix measured at 9.4 T.** MR images were acquired with a linear polarized birdcage and SNR was computed according to Method 2 described in [41].

The influence of signal attenuation on image quality for multiple nuclei was evaluated experimentally on the same NR system using an interchangeable set of quadrature birdcage coils with integrated transmit/receive switches and preamplifiers [42] and the prototype switch matrix shown in Fig 1 placed behind the coil connectors. Imaging of a phantom (50 ml cylindrical tube filled with doped water and 150 mM NaF (Sigma-Aldrich, German)) was carried out using the FLASH sequence at the $^1$H, $^{19}$F and $^{23}$Na frequencies. The sequence parameters were TE/TR = 3.95 ms/150 ms, FoV = 52x52 mm$^2$, Matrix size 256x256, slice thickness 0.2 mm with a total acquisition time of 38.4 ms for the proton experiments, TE/TR = 2.48 ms/450 ms, FoV = 42x42 mm$^2$, Matrix size 42x42, slice thickness 1 mm, flip angle 75˚, 16 averages with a total acquisition time of 5 minutes for $^{19}$F, and TE/TR = 2.85 ms/40 ms, FoV = 30x50 mm$^2$, Matrix size 30x50, slice thickness 1 mm, flip angle 60˚, 16 averages with a total acquisition time of 2:34 minutes (3D) for $^{23}$Na. The images acquired with and without the terminating capacitances present are shown in Fig 4.

As can be seen from Figs 3 and 4, small values of attenuation already have a negative influence on the SNR of the image. Therefore, the switch matrix should be optimized for as low IL as possible. Consequently, for larger switch matrix sizes, which are likely to be encountered in the modern MRI systems, an algorithm is required that determines optimum compensation element values in an automated way. This optimization problem can use a single cost function (C) and minimizes the maximum insertion loss (IL) encountered for all switch configurations

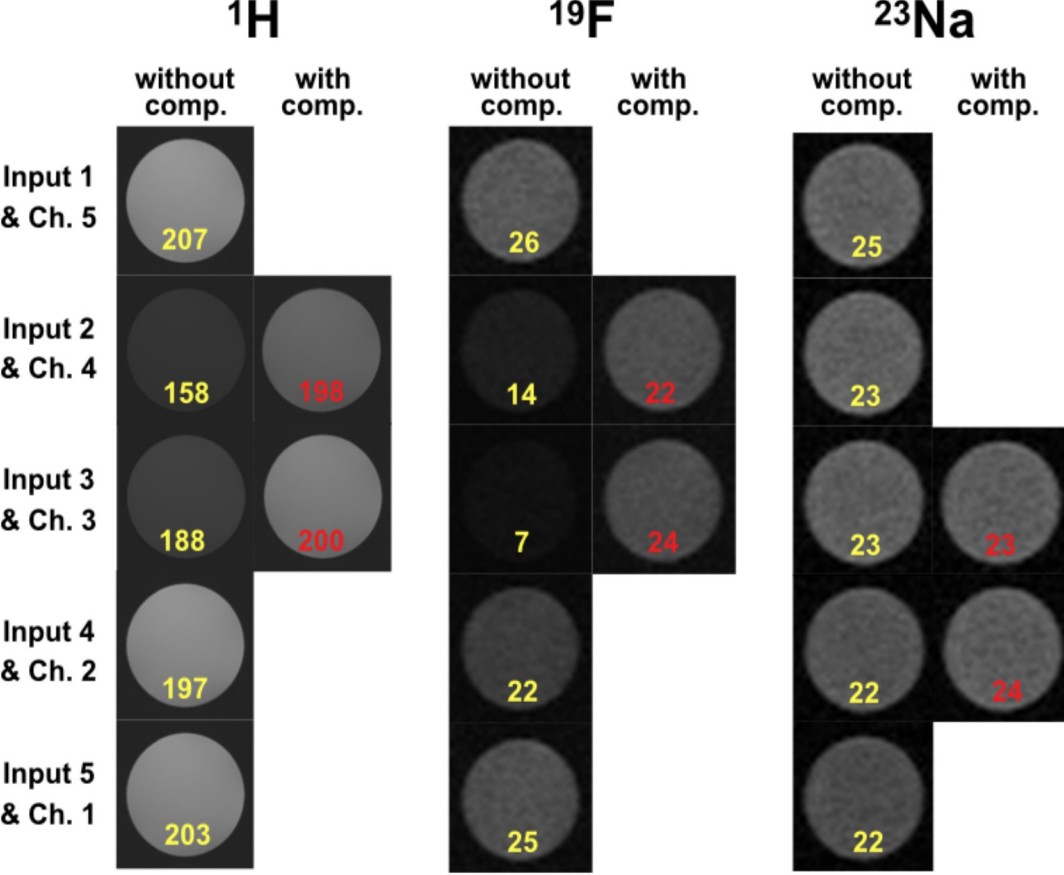

**Fig 4. Measured FLASH images for different nuclei when the prototype switch matrix is present in the 9.4 T animal MR system.** The yellow text gives SNR values when leaving transmission lines uncompensated, the red text shows compensated (91 pF) transmission line stubs.

given that the optimum stub termination is selected for a given signal routing scheme. For a M x N switch matrix this is expressed as

$$C = \max(IL(i, j, f_k)) \quad \forall i = 1..M, j = 1..N \tag{1}$$

with M and N being the number of input and output lines, respectively, and $f_k$ being the operating frequencies. Eq (1) is similar to using a weighted Tchebycheff model with weighting coefficients set to equal one as described in [43]. Due to the large number of variables equaling possible terminations for each stub line, metaheuristic approaches [44] are suitable for optimization of the switch matrix circuit. Metaheuristics can be classified as single solution approaches which act on a single candidate solution, e.g. simulated annealing [45] and variant local search methods, or population-based approaches [46]. The latter includes nature-inspired algorithms, such as evolutionary algorithms, particle swarm optimization [47], firefly algorithms [48] and artificial bee colonies [49]. While metaheuristic population-based approaches are not guaranteed to find the globally optimum solution, a recent comparison has shown that they are competitive with deterministic Lipschitz algorithms [50] in many respects. This, along with their relative ease of use, are the motivation for applying them to the switch matrix optimization for multi-nuclear, high-field MRI.

IL can be computed as the sum of the mismatch loss and transmission losses over the transmission lines used in the switch matrix, e.g. compare [51]. Mismatch loss is computed by first transforming the compensation impedance of the two stub lines to the active switch junction. This uses the impedance transformation formula for transmission lines of characteristic impedance $Z_0$ terminated with the impedance $Z_L$ along the stub line length l

$$Z_{\text{input}} = Z_0 \frac{Z_L + Z_0 \tanh \gamma l}{Z_0 + Z_L \tanh \gamma l} \tag{2}$$

with γ being the complex propagation constant of the transmission line [52].At this point, the parallel impedance of the stub lines and the 50 Ohm load from the receiver are computed and further transformed to the switch matrix input. This enables the impedance, and hence the mismatch loss at this point, to be calculated.

The microstrip transmission lines used in the calculations have been designed to display a characteristic impedance of 50 Ohms when built on an FR4 substrate with a thickness of 1.5 mm (relative dielectric constant $\varepsilon_r$ = 4.6, copper trace thickness t = 0.035 mm, substrate thickness h = 1.5 mm, microstrip trace width w = 2.725 mm, loss tangent of substrate tan δ = 0.022, surface roughness–rms deviation of the conductor surface from a plane–Rough = 0.055, and relative conductivity with respect to copper Rho = 1). The transmission line parameters were calculated online (http://mcalc.sourceforge.net) using a method which employs the formulae derived by Hammerstad and Jensen [53] and is conveniently summarized in e.g. [54].

## Implementation of circuit optimization

Circuit optimization was carried out with Python [55] using the platypus package [56]. The cost function is implemented in two subroutines–one computing the optimum termination and returning minimal insertion loss for a single switch junction, while the outer subroutine computes the maximum insertion loss over all junctions. The cost function is parameterized so as to allow the maximal insertion loss for a number of operating frequencies to be found, as well as having the choice to place more than one compensation element per transmission line stub. It should be noted that the range of termination elements allowed was limited from 300 nH inductive to 200 pF capacitive, which covers available lumped, high-frequency elements with suitable self-resonances and quality factors. This range was also mapped onto a real

valued space, between +1 and -1 for equidistant gridding during program execution to ensure that standardized initialization of the search space remained feasible. The final goal of the study was to design a multi-nuclear switch matrix with 128x64 channels. The dimensions were selected according to the maximum number of receiver channels available in commercial 7 T systems and the maximum number of receive elements in an experimental coil array described so far [57].I Initial investigations of algorithm performance were carried out on differently sized matrices for the sake of simplicity.

In an initial evaluation, several multi-objective optimization algorithms based on evolution strategies—a variant of evolutionary computing which is inspired by acting upon populations underlying variation and selection in generational loops [58]–were compared in order to evaluate their performance for the switch matrix optimization task. This was done for a 64x32 sized matrix operating at the proton frequency of 400 MHz and allowing two compensation elements for each stub. For this investigation, the transmission lines were assumed to be lossless. The strategies investigated were: a non-dominated sorting genetic algorithm II (NSGA-II) [59] and its more recent version NSGA-III [60], a covariance matrix adaptation evolution strategy (CMA-ES) [61], a generalized differential evolution (GDE3) [62], an indicator-based evolutionary algorithm (IBEA) [63], a multi-objective evolutionary algorithm based on decomposition (MOEA/D) [64], a multi-objective (OMOPSO) [65] and speed-constrained multi-objective particle swarm optimizer (SMPSO) [66], a strength Pareto evolutionary algorithm (SPEA2) [67] and an epsilon multi-objective evolutionary algorithm (ε-MOEA) [68]. In addition to using different optimization methods, the strategies employed also differ in their approaches to evaluating the Pareto optimality of the population. For example, the NSGA derivatives employ dominance depth, while SPEA uses dominance count/rank; a different approach is to use performance measures for the selection step, e.g. in IBEA [43]. For an in-depth discussion of the properties of the different algorithms, the reader is referred to standard textbooks, e.g. [43]. A list of active projects implementing multi-objective optimization in the Python programming language is given in [69]. All algorithms were executed with 10 random seeds and ran for a maximum of 2000 iterations.

Using the optimization algorithm selected during the initial evaluation, a multi-nuclear switch matrix with 16 inputs and 8 outputs was designed for operation at the $^1$H, $^{19}$F, $^{31}$P, $^{23}$Na, and $^{17}$O Larmor frequencies of a 9.4 T system. This time, attenuation on the microstrip lines was accounted for by using a complex propagation constant for each frequency. In addition, a variable, but unique, spacing between neighboring transmission lines in the range 10 mm to 60 mm was considered during the optimization to allow the circuit geometry to be optimized in combination with the compensation elements. The lower limit was imposed so as to avoid closely spaced lines with high coupling between neighboring lines. Compensation elements were allowed from a maximum of 300 nH inductive to 200 pF capacitive, as above. This more realistic design example was evaluated with the three best-performing algorithms found during the initial comparison.

Finally, a large sized multi-nuclear matrix with 128 input lines and 64 output lines was optimized using the same constraints as given above. This time, only the ε-MOEA algorithm was investigated as it provided the best results for the two test cases investigated beforehand. The number of algorithm iterations was increased until no further improvement in the overall cost could be obtained.

## Results

Table 1 shows the highest insertion loss for any path across a 64x32 sized matrix obtained from optimizing the compensation elements with the different algorithms. It can be seen that

**Table 1. Highest insertion loss (IL) obtained for any switch configuration for a 64x32 sized, proton only matrix when neglecting transmission line losses.**

| Optimization Algorithm | Highest IL for any signal path [dB] |
| --- | --- |
| NSGA-II | 1.198 |
| NSGA-III | 6.884 |
| CMA-ES | 4.335 |
| GDE3 | 5.041 |
| IBEA | 1.205 |
| MOEAD | 12.278 |
| OMOPSO | 3.617 |
| SMPSO | 2.109 |
| SPEA | 1.208 |
| ε-MOEA | 1.140 |

NSGA-II, SPEA and ε-MOEA performed best in this task, while other algorithms converged to significantly worse solutions, despite being executed for a number of seed values.

The three algorithms that converged to an insertion loss below around 1.2 dB were further evaluated on a more realistic multi-nuclear switch design task. This task was based on a 16 x 8 switch matrix and the insertion losses for each nucleus and each switch configuration are shown in S1 to S3 Figs. The overall results for the optimization with a maximum number of algorithm executions of 4000 times are given in Table 2.

The results for the full-sized switch matrix (128 x 64 ports) designed with ε-MOEA are given in Fig 5. Table 3 provides a detailed performance evaluation for different numbers of algorithm execution.

For the 128x64 matrix, a maximum IL over all nuclei of 3.81 dB was encountered. Horizontal and vertical transmission lines were spaced at 10.0 mm and 10.4 mm, respectively. Thus, the maximum transmission line length was approximately 1885 mm (123 x 10.0 mm + 63 x 10.4 mm), resulting in an attenuation of 3.28 dB at the proton frequency along the line. The high losses attributed to attenuation along the transmission lines become clearly visible in the lower-left corner (corresponding to the longest path of the switch matrix) of the $^{1}$H and $^{19}$F attenuation maps shown in Fig 5. The mean insertion losses and standard deviations of the optimized matrix were 2.38 dB ± 0.86 dB, 2.26 dB ± 0.81 dB, 1.19 dB ± 0.44 dB, 0.90 dB ± 0.36 dB and 0.63 dB ± 0.39 dB for the $^{1}$H, $^{19}$F, $^{31}$P, $^{23}$Na and $^{17}$O frequencies, respectively.

## Discussion

We have shown that SNRs of both, proton and X-nucleus signals can be significantly affected when using a switch matrix to route receive signals from the coil connectors on the patient table to the receivers of an MR system. Thus, suitable circuit topologies that reduce signal attenuation in the switch matrix as much as possible are required. In this manuscript we

**Table 2. Overall performance of the optimization algorithms for the 32 x 16 multi-nuclear switch matrix design (IL–insertion loss, TL–transmission line).**

| NSGA-II | SPEA2 | ε-MOEA |
| --- | --- | --- |
| Highest IL: 1.20 dB | Highest IL: 1.31 dB | Highest IL: 1.00 dB |
| Vert. TL distance: 10.2 mm | Vert. TL distance: 18.5 mm | Vert. TL distance: 10.7 mm |
| Hor. TL distance: 12.4 mm | Hor. TL distance: 10.9 mm | Hor. TL distance: 10.2 mm |
| Execution time: 263 s | Execution time: 335 s | Execution time: 1054 s |

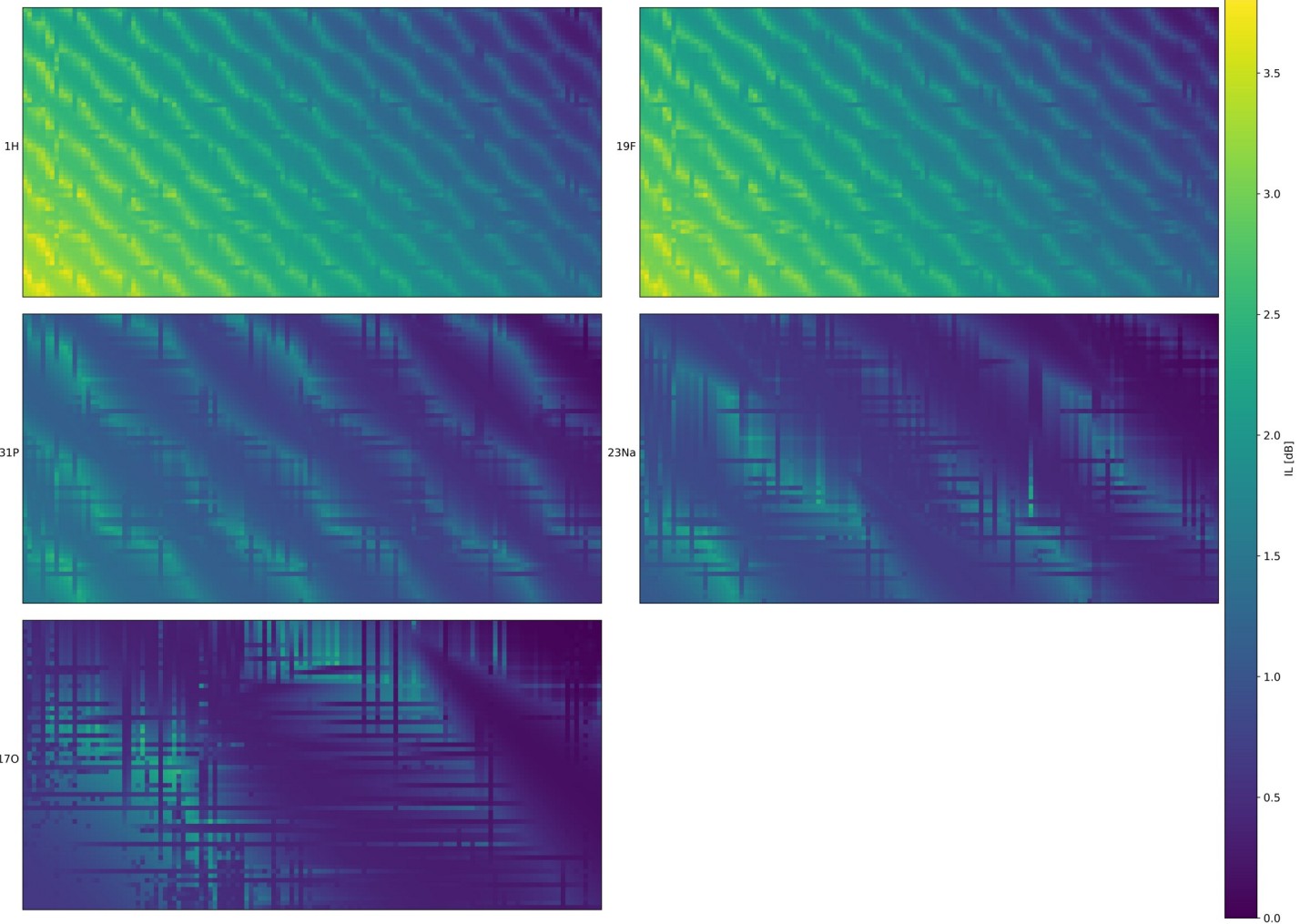

**Fig 5. Maximum IL for the full switch matrix size for all nuclei investigated as obtained with the ε-MOEA algorithm.** For better visibility, the switch matrix was rotated so that the output ports are shown on the left-hand side (abscissa) ranging from 1 (top) to 64 (bottom). The inputs are arranged along the ordinate and increase from right to left. Each color patch gives the maximum IL for the connection between the corresponding input and output port. All data are simulation results calculated using the transmission line formulas from Hammerstad et al. [53].

demonstrate that circuit optimization of a large, multi-nuclear switch matrix is feasible even when using a limited number of compensation elements to compensate for the influence of open-ended transmission lines of varying length. Optimization, although using a single cost function only, was carried out with a variety of multi-objective optimization strategies which employ either swarm like features or genetic principles and are thus suitable candidates for metaheuristic, non-convex problems. The non-convexity of the optimization problem at hand arises from the magnitude operation in the IL formulation. These algorithms have been

**Table 3. Performance as a function of the number of algorithm executions for the full-sized matrix design.**

| N = 2000 | N = 4000 | N = 8000 | N = 16000 |
|---|---|---|---|
| Highest IL: 5.06 dB | Highest IL: 4.25 dB | Highest IL: 3.84 dB | Highest IL: 3.81 dB |
| Vert. TL dist.: 11.2 mm | Vert. TL dist.: 10.4 mm | Vert. TL dist.: 10.0 mm | Vert. TL dist.: 10.0 mm |
| Hor. TL dist.: 23.3 mm | Hor. TL dist.: 13.0 mm | Hor. TL dist.: 10.2 mm | Hor. TL dist.: 10.4 mm |

employed successfully in other fields of MRI, e.g. for $B_1$ shimming [70] using different single cost functions and a particle swarm approach and in coil circuit optimization [71] with three optimization targets and employing CMA-ES. In the presented optimization task, the optimization was able to generate a circuit topology that caused an addition IL of only 0.6 dB above that of the attenuation of the longest transmission line. With a maximum overall IL of below 4 dB, the degradation in image SNR is negligible, as we demonstrated with attenuation measurements on a 9.4 T scanner.

In our investigation, we found ε-MOEA to perform best out of the tested candidate algorithms. While this is a lesser-known evolution strategy [72] and was shown to be less performant in an biobjective testbed [73] it consistently outperformed the other algorithms implemented in the platypus collection. The good performance of the ε-MOEA algorithm might be due to the fact that, in contrast to generational algorithms which evolve the entire population at every iteration, ε-dominance archives ensure convergence and diversity throughout search for a proper selection of the ε parameter [74]. However, further investigation is required to back up this hypothesis.

From initial tests using multiple cost functions–single cost functions for all switch matrix junctions–it quickly became evident that one combined target function performed superior. This is in line with [70], where the formulation of multiple optimization goals was found to be feasible but was not chosen in the final implementation. The number of algorithm executions for the optimization to converge of around 16,000 is in the range reported in [50]. While the metaheuristic approaches employed are not guaranteed to converge to the global optimum, a comparison with the IL on the longest transmission line arising from attenuation only lends reasonable credit only to the results obtained with the proposed optimization strategy.

It should be noted, therefore, that for the investigations presented here, no experimental validation has been carried out for the full-scale matrix, since the intention was solely to investigate a feasible solution of the given optimization problem. However, from the investigations carried out in prior work [26, 27], we feel that agreement between simulation and measurement has been sufficiently demonstrated. Further validation is given in S4 Fig which compares IL for a 6x3 matrix optimized with the proposed algorithm with those obtained from a circuit simulator. The impact on of insufficient transmission line compensation on MRI image quality has been investigated both previously and in this work and allows a compromise between the envisaged switch matrix implementation and the associated image degradation to be found.

While this work focuses on the classical topology of a PIN-diode based switch matrix, which results in a rather large PCB footprint–e.g. the 128 x 64 switch matrix designed in this work measures approximately 1250 mm x 660 mm–the workflow can be used in combination with alternative matrix implementations. One possibility could, for example, be to use the switch matrix designed in this work to optimize the sub-matrices presented in [27] for stacked matrix designs or to reduce the flexibility of the switch matrix by using a multiplexer circuit that combines a number of input lines prior to routing through the switch matrix, e.g. as suggested in [21, 25]. It is certainly also feasible to change the number of allowed compensation elements, e.g. using a single termination only as described previously or to increase the number of elements to three to decrease the maximum IL further.

## Supporting information

**S1 Fig. Maximum IL for all switch configurations for all nuclei investigated as obtained with the NSGA-II algorithm for a 16x8 sized matrix.** Note that the optimum termination for each case is given in the respective field for the vertical line (top row) and horizontal line

(bottom row).
(PNG)

**S2 Fig. Maximum IL for all switch configurations for all nuclei investigated as obtained with the SPEA2 algorithm for a 16x8 sized matrix.**
(PNG)

**S3 Fig. Maximum IL for all switch configurations for all nuclei investigated as obtained with the ε-MOEA algorithm for a 16x8 sized matrix.**
(PNG)

**S4 Fig. Comparison of IL computed with the proposed method and obtained by circuit simulation using the free software package QUCS (available from qucs.sourceforge.net).** Both plots are normalized with respect to their maximum IL in color coding. While the pattern looks similar the circuit simulator predicts slightly larger insertion losses in all cases, which is probably due to using slightly different formulae for calculating transmission line properties.
(PNG)

## Acknowledgments

The authors would like to thank the reviewers of [26] for bringing the idea of circuit optimization for large-scaled switch matrices to the attention of the authors. We sincerely hope that this work does justice to the questions raised during the review process. We thank C. Rick for careful proof reading of this manuscript.

## Author Contributions

**Conceptualization:** Jörg Felder, Chang-Hoon Choi, N. Jon Shah.

**Investigation:** Jörg Felder, Chang-Hoon Choi, Yunkyoung Ko.

**Methodology:** Jörg Felder.

**Project administration:** Jörg Felder.

**Software:** Jörg Felder.

**Supervision:** Jörg Felder, N. Jon Shah.

**Validation:** Yunkyoung Ko.

**Visualization:** Jörg Felder.

**Writing – original draft:** Jörg Felder, Chang-Hoon Choi, N. Jon Shah.

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
