## [Decision Letter · Decision Letter 0]

13 Feb 2020

PONE-D-19-28452

Optimization of high-channel count, switch matrices for multinuclear, high-field MRI

PLOS ONE

Dear Dr. Felder,

Thank you for submitting your manuscript to PLOS ONE. After careful consideration, we feel that it has merit but does not fully meet PLOS ONE’s publication criteria as it currently stands. Therefore, we invite you to submit a revised version of the manuscript that addresses the points raised during the review process.

We would appreciate receiving your revised manuscript by Mar 29 2020 11:59PM. To enhance the reproducibility of your results, we recommend that if applicable you deposit your laboratory protocols in protocols.io, where a protocol can be assigned its own identifier (DOI) such that it can be cited independently in the future. For instructions see: http://journals.plos.org/plosone/s/submission-guidelines#loc-laboratory-protocols

We look forward to receiving your revised manuscript.

Kind regards,

Cem M. Deniz

Academic Editor

PLOS ONE

Additional Editor Comments (if provided):

Your manuscript was reviewed by three experts in MRI system and coil design. All the reviewers identified that the study needs to be substantially improved in order to meet publication standards. Specifically, R1 requested you to incorporate more details of the method in the current manuscript. R2 suggested you demonstrate how does proposed switch matrices translate to MR image quality, and to extend the methods section to explain design parameters clearly. R3 suggested you improve the introduction by stating the motivation behind the work and identifying the problems of existing switches explicitly. As highlighted by the reviewers, I suggest you validate the simulations by providing an experimental proof-of-concept study as referencing to prior work (Ref. 19-20) would not be sufficient. In addition, the image quality of all figures needs to be improved. In the view of reviews and my assessment of the manuscript, I recommended a major revision to give you enough time to prepare the revised manuscript improved by a proof-of-concept study.

Journal Requirements:

2. We note that Figure 1 in your submission contain copyrighted images. All PLOS content is published under the Creative Commons Attribution License (CC BY 4.0), which means that the manuscript, images, and Supporting Information files will be freely available online, and any third party is permitted to access, download, copy, distribute, and use these materials in any way, even commercially, with proper attribution. For more information, see our copyright guidelines: http://journals.plos.org/plosone/s/licenses-and-copyright.

1.         You may seek permission from the original copyright holder of Figure(s) [#] to publish the content specifically under the CC BY 4.0 license.

"I have read the journal's policy and the authors of this manuscript have the following competing interests: NJS and JF are co-founders of Affinity Imaging GmbH a spin-off company that manufactures high field MRI coils for research purposes."

Reviewers' comments:

Reviewer's Responses to Questions

**Comments to the Author**

1. Is the manuscript technically sound, and do the data support the conclusions?

Reviewer #1: Yes

Reviewer #2: Partly

Reviewer #3: Partly

2. Has the statistical analysis been performed appropriately and rigorously? 

Reviewer #1: Yes

Reviewer #2: N/A

Reviewer #3: Yes

3. Have the authors made all data underlying the findings in their manuscript fully available?

Reviewer #1: Yes

Reviewer #2: No

Reviewer #3: No

4. Is the manuscript presented in an intelligible fashion and written in standard English?

Reviewer #1: Yes

Reviewer #2: Yes

Reviewer #3: Yes

5. Review Comments to the Author

Reviewer #1: This research paper provides a method of directly evaluating the loss in switch matrices used in receive arrays for different nuclei at 9.4T and compares different algorithms for finding terminations that minimize the insertion loss in the switching. The authors have very clearly outlined their process and detailed the results, while showing that significant improvements are obtained by the method. There is not enough details provided, and I am not specifically well informed enough, to determine if there are any issues with the implementation with the algorithms or if there was an expected result for the comparison of the different algorithms based on previous studies.

There are a number of specific comments on details and changes that can be made to improve the paper:

Page 3, line 55 – change “speed up” to acceleration factor

Page 4, line 80 – this phrasing makes it seem like multinuclear arrays are a new development, and the quotes reference is not a very early or specifically monumental reference of a multinuclear array. Please find more relevant and primary references and describe the development of X-nuclei array more fully.

Page 5, line 96 – replace “Note also that, both” with “The”

Figure 1- it may be the output settings for this pdf, but the figures are almost illegible, thus for publication will be necessary to make sure they are all of high quality

Figure 2- this graph/figure does well to highlight the importance of modifying the resulting stubs to avoid insertion losses that would degrade noise factor However it also seems to provide evidence that a more complicated approach is not needed, as the maximum loss of 2.21dB would have very little effect on the final noise figure considering the gain of the preamplifier stage.

The author’s previous publication “Signal Loss Compensation of RF Crossbar Switch Matrix System in Ultra-High Field MRI” also seems to point to this conclusion. This particular point should be addressed more explicitly.

Some rough estimate of at what point the losses may result in an SNR loss of perhaps >5% and 2% for a single channel would be valuable.

Page 8, line 175- add comma: “This time,”

Figures 3-5 do not provide much interesting information as there is no way to know from the text or discussion of the algorithm why specific switch combinations had lower insertion loss compared to others. However, there’s some value in seeing the difference with all the different nuclei at once. I suggest that this figure may only need to be shown for one algorithm, and then it can be detailed what specific differences the different algorithms had. Right now, it is not detailed why any specific differences are observed.

For this provided manuscript these figures 3-5 are also illegible

Page 12,line247- It would greatly enhance the work if some detailed explanation and discussion of why the �-MOEA algorithm performed best and where and why the other algorithms found non-optimal solutions. As this is the best performing algorithm it would also be best to detail it more explicitly in the methods.

Page 12,line256- I would agree that for the particular problem investigated here the theoretically derived insertion losses for transmission lines should match very closely with experimental.

Page 12, Line 264- The 1250 mm x 660 mm size is very large, A reference or details on what is currently used in commercial systems should be included. Also, in should be discussed: how necessary is it that every channel is able to be mapped or switched between every output. I believe this specific set-up is actually not very applicable for this reason.

Discussion points: why the different algorithms had different execution times and some idea of how much this could change with optimization.

Some description in the methods, if possible, of the specific classes the algorithms fall under

More time needs to be spent formatting the references correctly and more consistently.

Reviewer #2: The authors present a novel design methodology for a broadband switch matrix. Its frequency is ranging from 54 to 400 MHz. As it is seen in fig 1, it seems that its hardware does not require any novelty to make the design in broadband as the authors have published the design of the PIN-diode based switch matrix before (ref 19). The novelty in PIN-diode based switch matrix design is not clear in this manuscript. As the authors mentioned mostly in the introduction section on hardware design parameters, this manuscript is on various optimization algorithms employed to optimize the performance of the switch matric design. The choice of performance parameters demonstrated in this manuscript was not clearly explained in the methods section and its motivation was not clearly given in the introduction and discussion sections. The combination of design and optimization methods in the switch matrix was shown in the manuscript but its translation into the MRI image quality and image acquisition speed is missing in the manuscript.

In general, the used language in this manuscript is adequate but it contains a few grammatical errors. The relevant referencing was sometimes missing in the introduction. Unfortunately, the image quality of the figures in the pdf is very bad, so I have to ask to the authors to improve the quality of all figures in the manuscript. The tables require further explanations in the caption since the parameters and abbreviations were not included. My detailed comments and questions are attached in the pdf.

Reviewer #3: The authors compare several optimization methods for address the non-convex problem of optimizing RF PIN diode switch performance for MRI receive chains that have multi-nuclear capability. The authors show how adding a small number of discrete impedance elements to the end of stubs in a cross-bar switch network can greatly improve insertion loss by compensating for the variable lengths of stubs, which depend on which output a given input is connected to. The optimization method provides a design for 128 input ports and 64 output ports that works at several X-nuclei frequencies and provides maximum insertion losses that are at most 0.6 dB above the baseline loss for the transmission line in the switch itself. I think the strength of the paper lies in its comparison of several optimization methods that can handle non-convex objective functions, and in the comparatively simple switch network design that requires only four impedance options at the end of each stub to provide adequate degrees of freedom for achieving good impedance match across a variety of stub lengths and operating frequencies.

For general context, I think the authors should add a brief discussion in the introduction about how multi-nuclear signal reception is handled on existing ultra high field scanners that have X-nuclei capability. On these scanners you can choose which nucleus to measure on each receive channel on the patient table socket. Do the authors know what kind of switching hardware is used on these scanners? What about these existing switches is suboptimal? Too much insertion loss for certain configurations? A better discussion of the shortcomings of existing technology would help motivate the switch optimization framework described in the present manuscript.

In previous work the authors show some loss of image SNR due to transmission line losses (Ref. 19). How much does the switch insertion loss impact the final image SNR (and the total system noise figure)? The signals have already passed through a preamplifier, so subsequent losses have less impact on the overall noise figure of the system compared to losses at the coil itself before amplification.

Is there a closed-form expression for the insertion loss as a function of the stub lengths and termination impedances? This would be helpful for readers to see since it will provide more intuition for the optimization problem that is being solved. The authors reference “analytical transmission line formulas” but do not include them in the paper. Equation 1 is not particularly helpful by itself. I would include more detail here so that readers don't have to consult other publications to understand the objective function fully.

Some experimental results would help validate the proposed optimization framework. It would make the paper more convincing if the authors could show that a prototype switch circuit indeed works for all the multinuclear frequencies of interest with the predicted insertion loss in each channel. A small board (such as 5x5 channels) should be sufficient for proof of concept. This would significantly strengthen the manuscript. I don’t think actual NMR measurements are needed, just bench top validation using a network analyzer to verify that the predicted insertion loss at each frequency for each switch configuration matches the experimental results.

I would try to include more key results in the abstract. For example, what is the average insertion loss for a naive switch with no impedance compensation elements at the end of each stub, as compared to the optimized result that shows no more than 0.6 dB of added insertion loss above the basic transmission line losses.

3.28 dB of transmission line loss for the proton frequency seems quite high. Is a low-loss dielectric substrate used?

How will the switching between the different stub termination impedances be implemented in practice?

“In this context, designing a switch matrix suitable for operation with protons and the most common X-nuclei at high-field is desirable.” Can the authors clarify exactly what role the switch will play on a multi-nuclear scanner? For example using the same plug on the patient table to route different coil channels to different X-nucleus receiver channels inside the scanner.

Can the authors discuss why the problem is non-convex? What properties of the objective function make it non-convex?

I think it would be good to include more details about the circuit design from Ref. 19 for readers who don’t have time to consult that reference. For example, I found it helpful to look at Fig. 2c from Ref. 19. At a minimum, I would draw the PIN diode circuit symbol in at least one of the figures to show how it functions in a crossbar switch, to help readers appreciate why the stubs remain after an input is connected to a desired output channel.

Please expand caption for Figure 2 and add appropriate labels to figure. Explain explicitly what is meant by “Input 1, 2, 3…”, “Channel 1, 2, 3, …” and the letters “A, B, C, …”

Figure 1. Is “voltage controlled RF switch” the same as PIN diode switch? A PIN diode is really “current-controlled” since it requires current to be forward biased, so it might be more accurate to call it current-controlled or just a “PIN diode-actuated switch”.

Figure 2 caption. Please add more detail to the caption (and manuscript text), for example describing what is denoted with the red text and red outline. Also why aren’t the other switch cases considered, for example Input 1 going to Channels 1-4? Shouldn't all inputs be able to connect to all output channels? This is not clearly explained in the text. Are the values reported in Fig. 2 calculated or measured experimentally?

I would recommend adding more labels and details to Figures 3-5. Please add labels for the rows and columns of each matrix. And please add more detail to the caption explaining what is going on in the matrices/tables. I assume the two entries in each cell correspond to the selected impedance for the two stubs realized on the two sides of the PCB for minimizing the insertion loss. Same for Figure 6, please add labels to each panel in the figure. I assume the dimension of the arrays is 128x64? Please clarify that the values are simulated.

Is the 17O insertion loss generally lower than the other isotopes because of its lower Larmor frequency?

“…allowed was limited from 300 nH inductive to 200 pF capacitive, which covers available lumped, high frequency elements with suitable self-resonances and quality factors.” I assume these parts are non-magnetic and that this limits the readily available range of component values. Or were there other factors motivating the specific choice of impedance matching component values?

p. 7: “128x64 channels…” What does each dimension refer to? 128 inputs and 64 outputs?

Abstract

“…high field systems…”

“As multichannel X-nucleus coil arrays provide the same array SNR benefits as in proton imaging, there is compelling motivation to also use coil arrays in these SNR-starved applications.”

p. 4: “However, with the increased availability of ultra high field scanners, the benefits of X-nuclei (non-proton, resonating at different frequencies) imaging is likely to attract increased interest.”

p. 5: “…as previously described in Ref. (19) for the connections…”

p. 7: “tuple” is somewhat uncommon usage here.

p. 9: “…while other algorithms converged to significantly worse solutions…”

p. 9: “This task was based on a…”

p. 12: “…lends reasonable credibility only to the results obtained…”

6. PLOS authors have the option to publish the peer review history of their article (what does this mean?). If published, this will include your full peer review and any attached files.

Reviewer #1: Yes: Adam Maunder

Reviewer #2: No

Reviewer #3: Yes: Jason P. Stockmann

---

## [Author Response · Author response to Decision Letter 0]

28 May 2020

All responses to reviewer and editor comments are uploaded as a separate file.

---

## [Decision Letter · Decision Letter 1]

20 Jul 2020

PONE-D-19-28452R1

Optimization of high-channel count, switch matrices for multinuclear, high-field MRI

PLOS ONE

Dear Dr. Felder,

Thank you for submitting your manuscript to PLOS ONE. After careful consideration, we feel that it has merit but does not fully meet PLOS ONE’s publication criteria as it currently stands. Therefore, we invite you to submit a revised version of the manuscript that addresses the points raised during the review process.

We look forward to receiving your revised manuscript.

Kind regards,

Cem M. Deniz

Academic Editor

PLOS ONE

Additional Editor Comments (if provided):

Your manuscript received favorable feedback from reviewers. There are minor changes needs to be addressed as highlighted by Reviewers 1 and 3. Please prepare a revision addressing reviewers' suggestions.

Reviewers' comments:

Reviewer's Responses to Questions

**Comments to the Author**

1. If the authors have adequately addressed your comments raised in a previous round of review and you feel that this manuscript is now acceptable for publication, you may indicate that here to bypass the “Comments to the Author” section, enter your conflict of interest statement in the “Confidential to Editor” section, and submit your "Accept" recommendation.

Reviewer #1: All comments have been addressed

Reviewer #2: All comments have been addressed

Reviewer #3: All comments have been addressed

2. Is the manuscript technically sound, and do the data support the conclusions?

Reviewer #1: Yes

Reviewer #2: Yes

Reviewer #3: Yes

3. Has the statistical analysis been performed appropriately and rigorously? 

Reviewer #1: Yes

Reviewer #2: Yes

Reviewer #3: Yes

4. Have the authors made all data underlying the findings in their manuscript fully available?

Reviewer #1: Yes

Reviewer #2: Yes

Reviewer #3: Yes

5. Is the manuscript presented in an intelligible fashion and written in standard English?

Reviewer #1: Yes

Reviewer #2: Yes

Reviewer #3: Yes

6. Review Comments to the Author

Reviewer #1: I think the manuscript has been significantly improved. The information provided, with the evaluation of loss in SNR with different nuclei with different attenuation, puts into context the improvements in IL with the switch matrices. The additional literature references on the algorithms and availability of the used algorithm provide a way to evaluate the success of the implementation here as well. The comments listed below can be corrected in editing and don’t require another review.

Minor Comments:

In abstract,

Instead of writing “For a matrix with 128 inputs and 64 outputs a realization is proposed that

displays a worst-case insertion loss of just 0.6 dB above that of the attenuation of the longest transmission line.”

Write

“For a matrix with 128 inputs and 64 outputs a realization is proposed that

displays a worst-case insertion loss of {the insertion loss}”

page 3, row 92: have defined “single-pole doublethrow (SPDT) twice”

The introduction has been improved greatly and reads very well, with a more than extensive review of the literature.

The introduction has been improved greatly and reads very well, with a more than extensive review of the literature.

Page 11, row 256: units for surface roughness are missing, also which is being quoted, peak-to-valley roughness (Rz), average roughness (Ra), or RMS roughness (Rrms)?

Reviewer #2: Thanks for addressing all questions and comments.

Reviewer #3: Supporting Figure 4 has no labels accompanying the table/matrix. A label or title would be helpful.

For a PIN diode switch to be effective, a good bias tee must be used to inject the bias current into the PIN diode path, while blocking RF from the DC bias path. What kind of choke is used that is effective for blocking all the X-nuclei frequencies? Typically an inductor with appropriate self-resonance frequency a little bit higher than the Larmor frequency is used, but for the case of a switch handling multiple Larmor frequencies, this is more complicated. An inductor with suitably high inductance might work decently for all the frequencies used, but this might require a ferrite component. Some basic design details might be helpful to other investigators wishing to reproduce the work (using non-magnetic circuit components).

The authors write on p. 6, line 122, that on some systems, including 7T systems, "the switch matrix was removed and the signal was routed directly without the possibility to switch it to a different receiver unit". I'm still confused about this. On the Siemens Magnetom 7T at my institution, the coil file specifies whether the pins on Plug 3 should be used for, say, 1H or 31P signal reception. There is presumably a switch somewhere in the receive chain to route the signals accordingly.

7. PLOS authors have the option to publish the peer review history of their article (what does this mean?). If published, this will include your full peer review and any attached files.

Reviewer #1: **Yes: **Adam Maunder

Reviewer #2: **Yes: **Ozlem Ipek

Reviewer #3: **Yes: **Jason Stockmann

---

## [Author Response · Author response to Decision Letter 1]

27 Jul 2020

The detailed response to the reviewers' concerns has been uploaded as an additional file.

---

## [Editor Report · Decision Letter 2]

29 Jul 2020

Optimization of high-channel count, switch matrices for multinuclear, high-field MRI

PONE-D-19-28452R2

Dear Dr. Felder,

We’re pleased to inform you that your manuscript has been judged scientifically suitable for publication and will be formally accepted for publication once it meets all outstanding technical requirements.

Kind regards,

Cem M. Deniz

Academic Editor

PLOS ONE

Additional Editor Comments (optional):

Congratulations on the acceptance of your paper!
---

## [Editor Report · Acceptance letter]

5 Aug 2020

PONE-D-19-28452R2 

Optimization of high-channel count, switch matrices for multinuclear, high-field MRI 

Dear Dr. Felder:

I'm pleased to inform you that your manuscript has been deemed suitable for publication in PLOS ONE. Congratulations! Your manuscript is now with our production department. 

Kind regards, 

on behalf of

Dr. Cem M. Deniz 

Academic Editor

PLOS ONE